



# Exploring the causes of multicentury pluvials in the Altiplano with a climate modelling experiment

Ignacio A. Jara[1], Orlando Astudillo[2], Pablo Salinas[2], Limbert Torrez-Rodriguez[2,3], Nicolás Lampe[4], Antonio Maldonado[2,5]

[1]Departamento de Ciencias Históricas y Geográficas, Universidad de Tarapacá, Arica, Chile
[2]Centro de Estudios Avanzados en Zonas Áridas (CEAZA), Colina del Pino, La Serena, Chile
[3]Universidad de La Serena, La Serena, Chile
[4]Departamento de Ingeniería en Computación e Informática, Facultad de Ingeniería, Universidad de Tarapacá, Arica Chile
[5]Departamento de Biología Marina, Universidad Católica del Norte, Coquimbo, Chile

*Correspondence to*: Ignacio A. Jara (ijarap@academicos.uta.cl)

**Abstract.** Proxy records have long documented the existence of multicentury hydroclimate anomalies in the Altiplano of South America. However, the causes and mechanisms of these events are still largely unknown. Here we present the results of an innovative climate modelling experiment that explores the oceanic drivers and atmospheric mechanisms conducive to long-term precipitation variability in the southern part of the Altiplano (18-25°S). For doing so, we performed 100-yr-long hydroclimate simulations using a regional climate model forced by climatological conditions that resulted in pluvial December-January-February (DJF) seasons during historical times. Our modelling simulations produce long-term negative DJF precipitation trends for the southern Altiplano, suggesting that the mechanisms leading to historical wet summers are unable to sustain century-scale pluvials such as the ones documented in paleoclimate records. Our simulations show that permanent La Niña conditions, as well as SSTs anomalies in the southern tropical Atlantic, progressively reinforce upper and lower-troposphere features that inhibit moisture transport towards the Altiplano. We further observed increases in March-April-May precipitation, suggesting the emergence of a long-term seasonal shift. Our simulations reproduce a sustained northward migration of the Atlantic trade winds, resulting in contrasting hydroclimate responses between the Altiplano and the tropical Andes. The atmospheric processes associated with these differences provide a useful analogue for explaining divergences in proxy records. Our study shows how regional climate modelling can be used to test paleoclimate hypothesis, emphasizing the necessity of combining proxy and modelling data to improve our understanding of long-term hydroclimate change.



## 1 Introduction

Precipitation in the South American Altiplano is scarce and largely limited to the austral summer months (December, January, and February; hereafter DJF) (Fig. S1 in the Supplement), representing the main hydrological resource for the inhabitants of the high Andes of Bolivia, Perú, Chile, and Argentina (García et al., 2007; Canedo-Rosso et al., 2019). Moreover, DJF precipitation is critical for the sustainability of mountain glaciers, lakes, salt flats and wetland systems (Vuille et al., 2018; Satgé et al., 2019; Anderson et al., 2021); and the main driver of regional groundwater recharge (Blin et al., 2022). Hence, understanding the drivers and variability of hydroclimate change in the Altiplano is of primary socio-environmental interest. Interannual rainfall variability in the Altiplano is pronounced and results from a complex interplay between upper and lower-troposphere circulation features, which are ultimately modulated by sea surface temperatures (SSTs) in the equatorial Pacific (i.e., El Niño Southern Oscillation; ENSO) and Atlantic basins. At longer timescales, tree-ring chronologies indicate the existence of significant decadal and multidecadal hydroclimate changes linked to equatorial pacific SSTs (Crispín-Delacruz et al., 2022; Morales et al., 2012). However, much less is known about the drivers of precipitation at centennial timescales, which hampers a long-term perspective of historical variations and the outlining of future responses.

Holocene paleoclimate records have long documented the existence of widespread multicentury droughts and pluvials in the Altiplano and adjacent eastern and western cordilleras (e.g. Valero-Garcés et al., 1996; González-Pinilla et al., 2021; Apaéstegui et al., 2018). However, the causes and mechanisms of these long-term anomalies remain little-explored. For instance, recently published hydroclimate reconstructions based on pollen, isotope and dust fluxes indicate the existence of a ~300-yr pluvial event that occurred about 2000 year ago in the drier southern portion of the Altiplano (17-25°S; >3500 masl) (Jara et al., 2019; Jara et al., 2020; Kock et al., 2019; Hooper et al., 2020; Fig. 1). This humid event seems to be recorded in the central Peruvian Andes at 10°S (Kanner et al., 2013), although it is absent in records from the tropical Andes of Peru, Ecuador, and Colombia further north (e.g. Bird et al., 2011; Jaramillo et al., 2021; Haug et al., 2001). These spatial differences prompted the suggestion than an extra-tropical moisture source could have contributed to this event (Bird et al., 2011; Jara et al., 2019; Hooper et al., 2020). Yet, documenting the spatial distribution of past climate anomalies does not provide direct evidence of their underlying atmospheric mechanisms. Using climate model simulations offers an innovative approach to identify atmospheric processes that could have contributed to past hydroclimate events (Zhu et al., 2022; Orrison et al., 2022; Anderson et al., 2006).

Transient climate simulations forced by global circulation models are employed to reconstruct past trends and processes. However, they are usually implemented with a coarse spatial resolution, and therefore they fail to resolve the physical processes that control precipitation in regions with complex topographies (Ludwig et al., 2019; Cook and Vizy, 2006). This issue was recently raised by González-Pinilla et al. (2021), who showed that a globally forced transient simulation could not capture the full range of hydroclimate variability observed in Holocene records from the central Andes (14-24°S). Similarly, Orrison et





al. (2022) observed that centennial-scale precipitation changes in South America are underestimated in global climate models. For this study, we employed a regional model to simulate continental-scale atmospheric conditions. Our experiment aimed to test the role of Pacific and Atlantic SSTs anomalies in generating century-long pluvials in the southern Altiplano. We identified two different yearly ocean-atmosphere configurations that led to positive DJF precipitation anomalies during historical times and employed them as lateral boundary conditions in independent 100-year modelling runs. We analyzed the resulting precipitation trends, identified the underlying atmospheric mechanisms that caused long-term excursions, and discussed the implications for regional paleoclimate reconstructions. Our modelling experiment allowed us to address the following questions: Can SSTs anomalies in the Pacific and/or Atlantic basin drive centennial-scale precipitation changes in the southern Altiplano? And what are the atmospheric mechanisms responsible for the existence of extended pluvial events in this region?

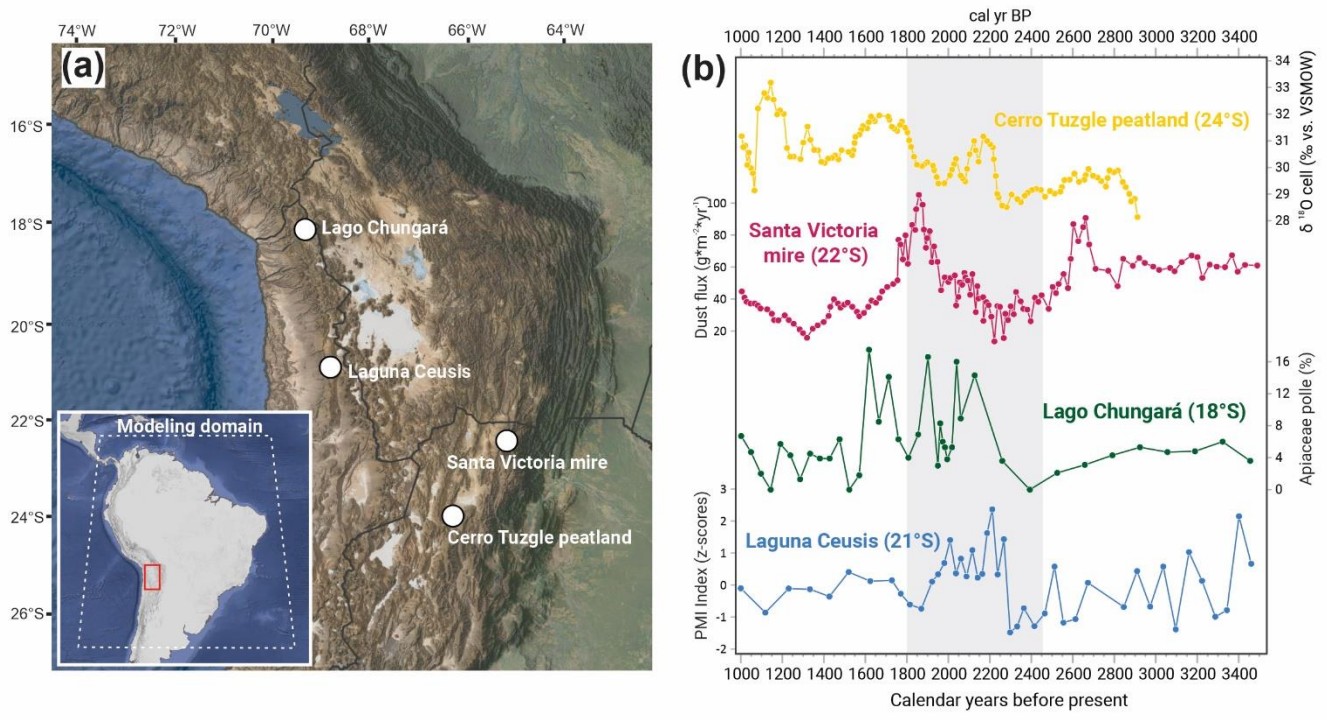

**Figure 1. (a) Map of the southern Altiplano, including the location of the paleohydroclimate records discussed in the text: Laguna Ceusis (Jara et al 2020), Lago Chungará (Jara et al., 2019), Santa Victoria mire (Hooper et al 2020) and Cerro Tuzgle (Kock et al 2019). The South American map inserted in the lower left corner shows the modelling domain and the southern Altiplano region (red rectangle). (b) Holocene hydroclimate records from the southern Altiplano showing a multi-century pluvial event about 2000 calendar year before presents (grey shading).**



## 2 Climate setting

The southern Altiplano experiences an extremely seasonal hydroclimate regime, with a DJF precipitation peak that represents about 70-80% of the annual budget (Fig. S1). DJF precipitation -rain and snow- results from the advection of continental moisture associated with the South American Monsoon System (SASM; Vera et al., 2006). The monsoon season is marked by the formation of a lower-troposphere (>800 hPa) Atlantic gyre, which transports oceanic moisture into the Amazon basin. On the western Amazon, the Atlantic moisture flux is redirected southwards by the Andes cordillera, forming an intense northerly flow of moisture referred in the literature as the South American Low-Level Jet (SALLJ; Romatschke and Houze Jr, 2010; Yabra et al., 2022; Fig. S2a). In addition, a zone of maximum cloudiness and precipitation is formed from the core of Amazon basin to the Atlantic coast of southeastern Brazil, which is referred to as the South Atlantic Convergence Zone (SACZ) (Kodama, 1992; Nielsen et al., 2019). The SACZ is one of the main features of the SASM, representing a pathway of moisture back to the Atlantic. Summertime advection of moisture to the Altiplano is promoted by the formation of the Bolivian High (BH), an upper-troposphere (< 500 hPa) anticyclonic circulation centered aloft the central Andes (Fig. S2b). The BH formation is linked to the strong monsoonal heating over the Amazon, the central Andes, and the SACZ regions (Lenters and Cook, 1997); and generates an upper-level easterly flow which promotes the transport of moisture up to the Altiplano (Garreaud et al., 2003). Yet, further away from the tropics, the southern Altiplano is at the periphery of the SASM domain, and therefore it receives considerably less moisture than the northern and eastern margins of the Andean plateau (Vuille and Keimig, 2004). By the end of the austral summer, the SASM and BH degenerate and upper-level westerly flow prevails over the Altiplano for the rest of the year.

Wet DJF seasons are usually associated with upper-level easterly wind anomalies, which are accentuated when the BH is displaced to the south (Aceituno and Montecinos, 1993). Expectedly, there is a strong correlation between DJF precipitation and upper-troposphere zonal flow in historical times (Fig. S3a). In addition, anomalous wet summers result from elevated moisture content in western Amazonia and/or from enhanced orographic lifting of the SALLJ (Segura et al., 2020). ENSO variability is an important driver of interannual DJF variability in the southern Altiplano, largely by modulating upper-level zonal wind anomalies (Sulca et al., 2016; Rojas-Murillo et al., 2022; Lavado-Casimiro and Espinoza, 2014; Vuille et al., 2000). The linkage between ENSO and the hydroclimate of the Altiplano is clearly revealed by the characteristic ENSO tongue that emerges from the spatial correlation between interannual DJF precipitation and SSTs (Fig. S3b). In general, wet summers are observed during cold SSTs in the tropical Pacific (La Niña) and vice versa (Garreaud et al., 2009; Vuille et al 2000). Atlantic SSTs, on the other hand, influence the strength of lower-level trade winds and moisture influx over the Amazon and the tropical Andes (Garreaud et al., 2003). In general, warmer northern tropical Atlantic (0-30°N) SSTs promote moisture transport into the Amazon basin (Segura et al., 2020). By contrast, warmer SSTs in the southern tropical Atlantic (0-30°S) are commonly associated with a weakened lower-level Atlantic gyre, reduced inland moisture transport, and drier Altiplano summers (Nielsen et al., 2019). DJF precipitation also exhibits decadal variability; however, the amplitude of these changes is smaller compared with the inter-annual oscillations (Garreaud et al., 2009). Decadal trends are associated with long-term variations in tropical



Pacific SSTs, with upper-level zonal wind anomalies resulting from changes in the position of the BH, and with strengthening/weakening of the SALLJ (Segura et al., 2016; Vuille et al., 2000; Sulca et al., 2022). Seiler et al. (2013) showed that dry (wet) summers in the southern Altiplano are associated with the positive (negative) phase of the Pacific Decadal Oscillation (PDO). There is no clear evidence for a strong influence of Atlantic SSTs over decadal rainfall variability in the Altiplano (Grimm and Saboia, 2015).

## 3 Methods

### 3.1 Selection of boundary conditions

We used Empirical Orthogonal Function (EOF) analysis on the ERA5 reanalysis dataset (Hersbach et al., 2020) to calculate the leading mode of DJF rainfall variability and identify anomalous wet summers over the 1951-2021 period (Fig. S4). Our modelling experiment focused on two summers with strong positive anomalies, albeit distinct oceanic and atmospheric configurations: the 1983/1984 and 2011/2012 DJF seasons (Fig. 2 and Table 1). Satellite and instrumental data confirm that these historical summers featured above-mean precipitation in the Altiplano (Fig. S4). Pluvial conditions were particularly intense over the 1983/1984 DJF season, associated with warm SSTs anomalies over the southeastern Pacific, the southern tropical Atlantic, and offshore the Brazil- Uruguay-Argentine coast down to 50°S (Fig. 2a). This pattern resembles the positive phase of the South Atlantic Ocean Dipole (SAOD; Nnamchi et al., 2011) (Table 1). In addition, the 1983/1984 summer featured a southwesterly located BH, fuelled by the formation of an upper-level low-pressure cyclone over the warmer southern tropical Atlantic (Fig. S5a). As a result, strong upper-troposphere easterly anomalies and moisture transport are observed over the Altiplano and further south in the Andes down to 30°S. By contrast, the 2011/2012 humid DJF season was associated with cold SSTs anomalies in the western and central equatorial Pacific (La Niña conditions), as well as strong cold anomalies over the southern tropical Atlantic, reminiscent of a negative SAOD phase (Fig. 2c and Table 1). Pluvial conditions in the Altiplano during the 2011/2012 summer were not related to a notable shift of the BH, but rather due to enhanced Atlantic moisture and upward motion over the western Amazon region, as well as intensified low-level transport to the central Andes (Marengo et al., 2013; Segura et al., 2020).

### 3.2 Modelling experiment

We evaluate if the SST anomalies that led to the 1983/1984 and 2011/2012 humid DJF seasons can sustain a century-long pluvial event in the southern Altiplano. For doing so, we conducted 100-yr climate simulations forced by replicating, year by year, the 3-hr SST fields and climatological conditions that produced those two humid summers. In addition, we used the 2003/2004 DJF season -a neutral hydroclimate year- to generate a control simulation run (Fig. 2b). This neutral simulation allowed us to evaluate to what extent the observed trends are developed in response to the model internal variability. A detailed list of start and end times of all modelling runs can be seen in Table S1. We used the regional Weather Research and Forecasting (WRF) model version 3.0 (Skamarock et al., 2008), with a continental-scale domain that included tropical and subtropical





South America, and the eastern Pacific and western Atlantic margins (90-30°W; 15°N-40°S; Fig. 1). We limited our simulation

to 100 years because of the high computational resources demanded for extended simulation periods in large domains. The

WRF runs were conducted on a 55 km horizontal grid spacing and 43 vertical levels, ranging from the surface up to 50 hPa,

with hourly instantaneous results. We used ERA5 reanalysis for the initial and lateral boundary conditions because this climate

dataset has proven to be an accurate depiction of temperature and precipitation in the Altiplano (Birkel et al., 2022).


**Figure 2. Hydroclimate and oceanic anomalies (z-scores) during the DJF periods selected for the climate simulation experiment. (a) precipitation and SST anomalies for the 1983/1984 summer. (b) same as (a) for the 2003/2004 summer (the selected control run). (c) same as (a) for the 2011/2012 summer. All anomalies are calculated as standard deviations from the 1951-2021 mean (z-scores).**






In terms of parameterizations, we followed the scheme adopted by Schumacher et al. (2020). To do so, we used the WRF Single Moment 6-class (WSM6, Hong et al., 2010) in physics. The radiation schemes followed the Rapid Radiative Transfer Model (RRTM, Mlawer et al., 1997) for longwave, and the Dudhia scheme (Dudhia, 1989) for shortwave was used. The MM5 similarity surface layer scheme was used (Berg and Zhong, 2005) and the thermal diffusion scheme was employed for land

surface. This land surface scheme is a Land Surface Model based on the MM5 5-layer soil temperature model with an energy budget that includes radiation, sensible, and latent heat flux (Huang et al., 2014). The Yonsei University parameterization of the planetary Boundary Layer (Hong et al., 2010) and the Kain-Fritsch scheme to atmospheric convection (Kain, 2004) were also utilized in our modeling runs. These parameterizations are identical to those in Schumacher et al. (2020) and showed a better performance when compared to the other schemes in the high complex topography of the Andes. We reported total DJF

precipitation trends in the 1983/1984 and 2011/2012 simulations plus the control run, to then analyze the simulated lower and upper-troposphere circulation and moisture transport at local and continental scales.

| | Precipitation anomaly (z-score) | El Niño 3.4 index | Souteastern Pacific SST anomaly (Figure 2) | South Atlantic Ocean Dipole Index (SAODI) |
|---|---|---|---|---|
| 1983/1983 DJF season | 2.4 | -0.6 | Warm | 0.5 |
| 2003/2004 DJF season | 0.23 | 0.4 | Cold | 0.63 |
| 2011/2012 DJF season | 1.7 | -0.8 | Warm | -0.68 |

**Table 1. Summary of precipitation and SST anomalies during DJF seasons selected as boundary conditions for the regional**
**simulations. El Niño 3.4 index was obtained from the NCAR (Climate Center for Atmospheric Research) Climate Data Guide. The SAODI index was based on the calculation presented by Nnamchi et al. (2011). For more information see section Code and Data availability.**

## 4 Results

For model validation, we compared the spatial precipitation variations during the first summer of the 1983/1984 and 2011/2012
WRF simulations, and their corresponding DJF season in the ERA5 and the satellite-rain gauge precipitation CHIRPS dataset (Funk et al., 2015; Fig. S6). At a continental scale, this modelling-historical comparison shows an overall good agreement between the spatial pattern of the WRF simulations and both climate datasets over the 1983/1984 and 2011/2012 summers, with precipitation maxima over the Amazon basin, the eastern Andean flanks, and the SACZ region. However, the WRF model simulates significantly higher precipitation over these regions in both simulations. Simulations with regional models at scales

of >50 km tend to overestimate rainfall, especially when it comes to convective precipitation (Torrez-Rodriguez et al., 2023). This is due to the fact that precipitation is not explicitly resolved in models at this scale, and because of the loss of spatial gradients imposed by the topography. Over the southern Altiplano, both WRF simulations and the ERA5 reanalysis show a similar spatial pattern of precipitation (although again, higher values are observed in the WRF), with drier conditions over the western and southern margins of the Andean plateau; whereas the CHIRPS dataset shows slightly lower precipitation amount





for the entire region. Notwithstanding these differences, the WRF model seems to have the ability to properly represent the
       main spatial precipitation patterns at continental and regional scales. Fig. 3 shows the total DJF precipitation trends for the
       southern Altiplano in our three 100-yr modelling runs. Long-term negative trends are visible for both 1983/1984 and 2011/2012
       simulations, as well as over our control 2002/2003 run. The 1983/1984 run shows the strongest downturn with a reduction of
       75% by the end of the simulation period, whereas this reduction is less intense (71%) for the 2011/2012 simulation. Our control
run shows 66% less DJF precipitation at the end of the simulation. Similar downward trends in all three modelling runs suggest
       that the simulated changes are not driven by the lateral boundary conditions, and that they might be controlled by the model
       internal variability or imposed by our experimental design, including the selected WRF parametrizations. In this regard, one
       potential explanation is the large size of the model domain employed in our study. In continental-scale domains, the lateral
       boundary forcing becomes less important than the internal dynamics and feedback (Caldwell et al., 2009). This could explain
why downward drifts are occurring in all three simulations even though they are forced by different boundary conditions. The
       common decline in DJF precipitation could have also resulted from the short spin-up period used in our simulation (10 days;
       Table S1), as longer spin-up times tend to reduce drift (Covey et al., 2006). However, the mechanisms responsible for the
       negative trends are different for each simulation (Figure 4; see next section), which indicates that our modelling results can
       reliably be used to explore the drivers of century-long hydroclimate anomalies in the southern Altiplano. We further note that
the negative drifts in our simulations are limited to the austral summer (Fig. S7), and that average annual precipitation for the
       entire South American continent show not common trajectories but distinct responses for each simulation (Fig. S8). In fact,
       positive trends are observed in certain parts of the simulation domain (Fig. 5), as will be discussed in the next section. All this
       evidence suggests that the external forcing is generating genuine climate features across the spatial domain, which largely
       result from the internal model dynamics and not from the selected WRF parametrization.


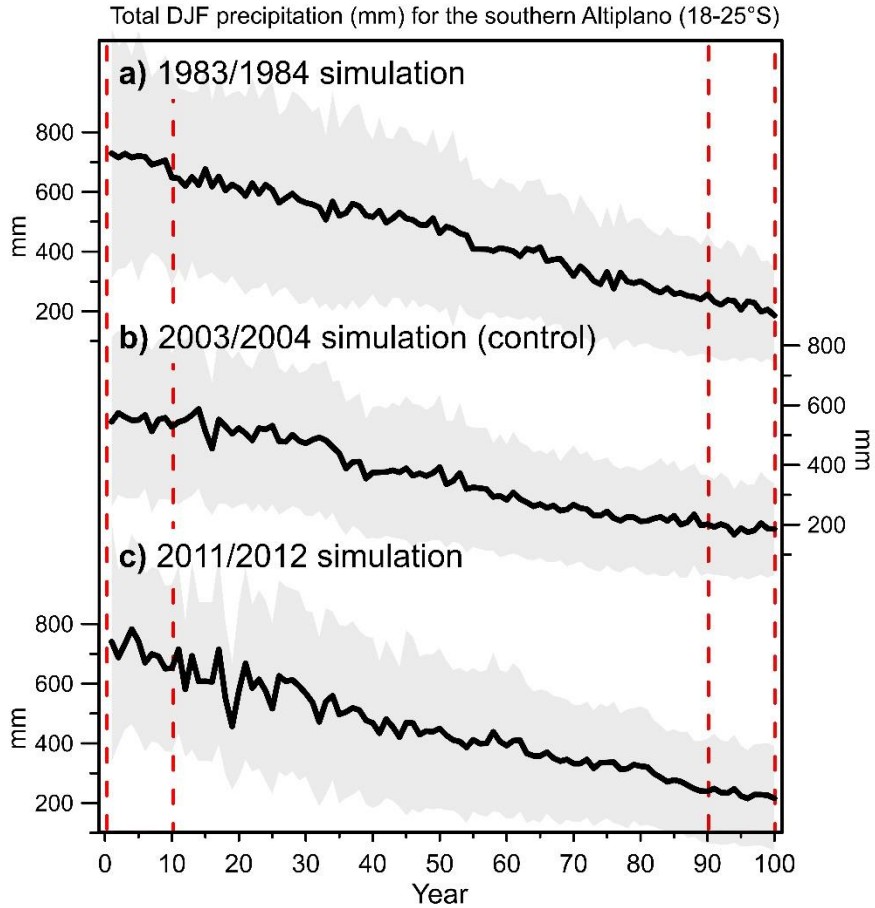

**Figure 3. Total summertime precipitation averages (mm) for the southern Altiplano region simulated in our three 100-yr WRF runs. The grey shading encompasses one standard deviation from the regional averages. The red rectangles denote the intimal and finale decades of the modelling runs which are discussed in section 5.**

## 5 Discussion and Conclusion

The causes and mechanisms behind the precipitation trends observed in the 1983/1984 and 2011/2012 simulations are explored in Fig. 4, which compares the averaged climatology over the initial (years 1 to 10) and final (years 91 to 100) decades in both simulations. For the 1983/1984 WRF run, the initial decade exhibits the typical upper and lower-level features that promote intense moisture transport towards the Altiplano (Fig. 4a). By contrast, the final decade exhibits a northward shift of the BH along with strong reduction in upward motion and specific humidity over the Andean plateau (Fig. 4b).



**Figure 4.** Comparative panel depicting the averaged atmospheric and moisture circulation during the initial and final decades of the WRF simulations. (a) Initial decade of the 1983/1984 simulation. Left panel: vertical motion at 500 hPa (colored) along with composite upper-level (200 hPa) zonal and meridional winds (streamlines). Upward (downward) motion is depicted as negative blue (positive red) values. Center: specific humidity at 500 hPa (colored) along with composite lower-level (800 hPa) zonal and meridional winds (streamlines). Right: pressure (hPa)-latitude cross section with anomalies (z-scores) for specific humidity (colored), integrated vertical and meridional circulation (vectors), and zonal winds (contours). The longitudinal means for all variables are calculated over the region delimited by the red rectangles in the center panels. Anomalies in the pressure-latitude sections were calculated as standard deviations from the 100-yr means. Continuous (dashed) contours indicate westerly (easterly) zonal wind anomalies. The red lines represent the profile of maximum elevation of the Andes cordillera and the position of the Altiplano is marked. (b) Same as (a) for the last decade of the 1983/1984 simulation. (c) Same as (a) for the first decade of the 2011/2012 simulation. (d) Same as (a) for the last decade of the 2011/2012 simulation.



Comparing the latitude/pressure plots for the initial and final decades of the 1983/1984 simulation reveals a change from easterly to a westerly wind anomaly aloft the Altiplano, as well as a transition from positive to negative anomalies in specific humidity at mid and upper levels (right panels in Fig. 4a-b). Moderate increases in upward motion and specific humidity are observed over the SACZ region during the final decade of the 1983/1984 run (Fig. 4b). Enhanced convection over the SACZ region might contribute to the negative precipitation trends, as it diverts the SALLJ back to the southern tropical Atlantic, and

thus away from the central Andes (Fig. 4b, central panel). A similar response is observed in the 2003/2004 control simulation, although in this later case the increase in upward motion is centred over southern Brazil and La Plata basin (Fig. S9). Moreover, any remaining moisture reaching the Andes is further diverted from the Altiplano by the prevailing upper-level westerly wind anomaly. Altogether, the implication of these results is that the SSTs anomalies responsible for the strong pluvial of December of 1983 and January-February of 1984– that is, warmer SSTs over the southern tropical Atlantic and the Atlantic coast of

southern Brazil, Uruguay, and Argentina- are unable to generate extended pluvial conditions in the region. We do not observe any evidence of extra-tropical moisture transport to the Altiplano by the end of the simulation (Fig. 4b; central panel). The hydroclimate of the Altiplano has been linked to the location/intensity of the SACZ at different timescales (Lenters and Cook, 1999). Our results are consistent with the observation that a warmer southern tropical Atlantic can sustain a vigorous SACZ over extended periods of time (Chaves and Nobre, 2004). However, it would not generate a century-scale pluvial in the

southern Altiplano if westerly wind anomalies prevail in the upper-troposphere. The combination of an intensified SACZ and a southward movement of the BH has been suggested as a driver for millennial-scale pluvial events over the Altiplano during the Last Glacial Termination (Martin et al., 2018). Differences with our modelling run may suggest that the interplay between the BH and the SACZ, and the resulting hydroclimate impacts on the Altiplano, may have not been stationary in time.

Comparing the initial and final decades of the 2011/2012 simulation also reveals a northward displaced BH and reduced specific humidity in the Altiplano, although we do not observe a strong reduction in upward motion (Fig. 4c-d). Unlike the 1983/1984 run, the final decade of the 2011/2012 simulation shows a decline in upward motion around the SACZ area, indicating a weakening of this climate feature (Fig. 4d, left panel). The primary cause of the precipitation decay in the 2011/2012 simulation could be related to a significant decrease in moisture availability over the western Amazon and its

southward transport by the SALLJ, as it is evidenced in the vertical motion and humidity plots (Fig. 4d, left and middle panels). The latitude-pressure plot of fig. 4c-d (right panels) reveals a transition from an easterly to a westerly zonal wind anomaly above the Altiplano, which further prevents any tropical moisture from penetrating the high Andean plateau. Overall, these results show that the SSTs anomalies causing the historical 2011/2012 DJF pluvial -that is, La Niña conditions over the tropical Pacific and cold temperatures in the southern tropical Atlantic- are incapable of forcing a century-long pluvial event.

Regardless of any change in conductivity and moisture availability, strong upper-level westerly wind anomalies are observed by the end of our three WRF simulations, which emphasizes the relevance of high-tropospheric flow in controlling century-scale hydroclimate change in the southern Altiplano. Notably, Atlantic SST anomalies observed during the 2011/2012 summer resemble the negative phase of the SAOD, with colder SSTs anomalies in the subtropics and warmer anomalies in the southern





Atlantic. In historical times, this configuration tends to weaken the lower-troposphere Atlantic gyre and the inland moisture
transport, generating dry conditions in the central Andes and northern Brazil (Chaves and Nobre, 2004; De Mendonça Silva et
al., 2023). Our simulation indicates that this mechanism might act cumulatively in our cyclical simulations, reducing the inland
moisture transport and SACZ activity over time. We further note that, by the end of all three simulations, the Atlantic branch
of ITCZ has shifted northward to 5-15°N (center panels of Fig. 4b and 4d). Such a northerly position is unusual in the austral
summer, and it is ultimately responsible for declining humidity levels and convective activity over the Amazon and the central
Andes. By contrast, strong positive precipitation anomalies are seen over the tropical Andes north of the Equator (Fig. 5). In
the case of 2011/2012 simulation, the ITCZ shift can be explained by the moderate warm SST anomalies observed over the
Atlantic coast of northern South America (5-20°N; Fig. 2c) during the 2011/2012 summer, which might fuel convection
progressively over time. Interestingly, we observed a long-term (70-80 year) increase (~50-60%) in total March, April, May
(MAM) precipitation in both simulations (Fig. S7). This is the only season experiencing significant precipitation surges in both
simulations and, by the end of the 100-yr runs, total DJF and MAM precipitation are roughly equivalent in the 1983/1984 and
2011/2012 runs. However, we note that these MAM surges are insufficient to drive significant total annual precipitation
increases in both simulations.

Our WRF simulations suggest that the oceanic conditions conducive to historical DJF pluvials in the southern Altiplano fail
to generate the extended periods of increased precipitation documented in Holocene records, including the one recorded ~2000
years ago. The accumulative effect of historical SST and atmospheric forcings in the WRF runs results in lower and upper-
troposphere alterations that, in contrast, lead to significant negative precipitation anomalies. Which SSTs anomalies drove the
multicentury pluvials observed in Holocene proxy records, then? Extended intervals with strong La Niña conditions might be
a potential cause, as we only tested neutral (1983/1984) and weak La Niña (2011/2012) summers (Table 1 and Fig. 2). We
note that one of the alkenone-derived SSTs reconstructions presented in Salvatteci et al. (2019) exhibits a cooling event in the
Pacific coast of southern Perú (15°S) between 2300 and 1900 calendar years BP, in close alignment with the southern Altiplano
pluvial event described in the introduction. This may suggest that a cooling of the southwestern Pacific coast could be a
potential SSTs driver; however, both 1983/1983 and 2011/2012 simulations were forced by warm SSTs over the Chile-Peru
coast (Fig. 2). Future modelling experiments using a broader set of oceanic forcing configurations will help to evaluate this
possibility.

At continental scale, both WRF runs exhibit strong negative trends over the Altiplano and central-northern Brazil, albeit
precipitation surges are observed over the western fringes of the Amazon, the tropical Andes, and northern South America
(Fig. 5). Opposite hydroclimate trends between paleoclimate records from the Altiplano and the tropical Andes have been
described in the literature (e.g., Apaéstegui et al., 2018; Jara et al., 2020). For instance, during the Little Ice Age (500-100 cal
yr BP; LIA) humid conditions are recorded in the tropical Andes of Peru, Ecuador, and Colombia (e.g., Bird et al., 2011; Vuille
et al., 2012; Kanner et al., 2013), while dryer climates have been suggested for the southern Altiplano and the Atacama desert

(e.g., Valero-Garcés et al., 1996; González-Pinilla et al., 2021; Mujica et al., 2015). A northward migration of the BH and the trade winds offer a potential mechanism to explain such differences, although these findings should be approached with care

because that proposed LIA scenario differs from the historical hydroclimate conditions tested in this study. We also note that the continental hydroclimate trends observed in our simulations, particularly the 1983/1984 run, resemble the second mode of past SASM variations identified by Campos et al. (2019) based on oxygen-isotope records. This mode of variability tracks SASM contractions/expansions that result in precipitation anomalies of opposite signs along its northern and southern margins. Such monsoon dynamics results in drier conditions over the southern Altiplano and northern Brazil, along with wet climates

over the tropical Andes and northern South America. Our simulations provide a potential mechanism for this type of monsoon behavior, which in Campos et al. (2019) dates between 460 and 300 cal yr BP.

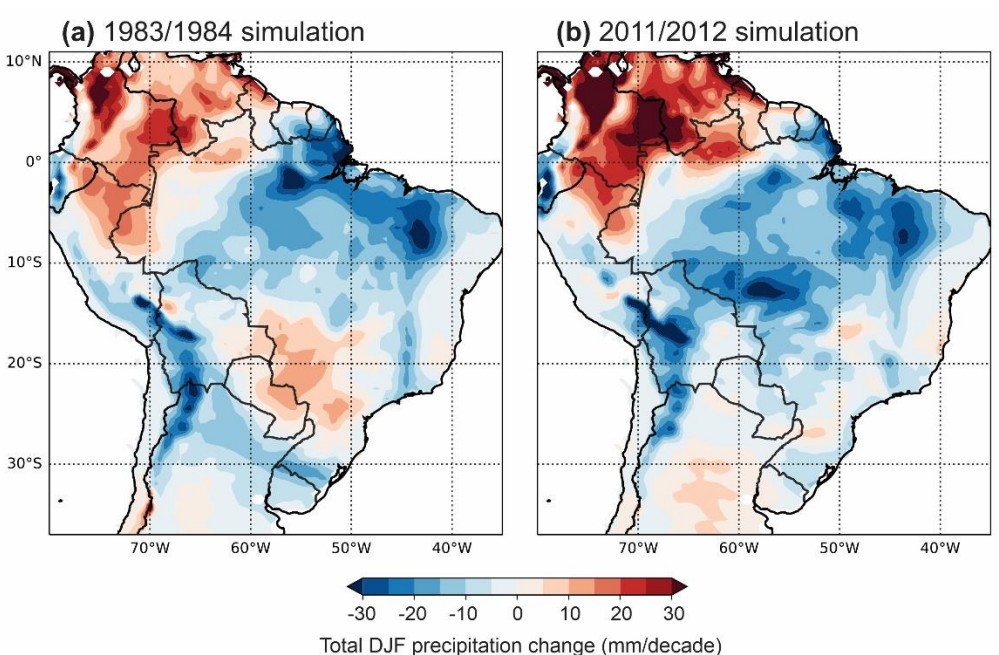

**Figure 5. Total DJF precipitation trends (mm per decade) in South America for the 1983/1984 and 2011/2012 simulations. Trends**
**are calculated using a linear regression for the entire 100-yr simulation period.**

To our understanding, this study presents the first regional climate modelling experiment of long-term hydroclimate responses in the southern Altiplano, yielding novel information regarding the mechanisms that govern precipitation variability at

centennial timescales. Although we could not provide direct evidence for the causes of the multicentury pluvials recorded in proxy records, we presented an outline of potential oceanic and atmospheric drivers explaining long-term hydroclimate trajectories. Since summertime precipitation is a critical socio-environmental resource for the inhabitants of the high Andes of South America, our results may help to outline future hydroclimatic trajectories in a region where climate projections show considerable disagreement (Minvielle and Garreaud, 2011; Segura et al., 2020; Vera et al., 2019). Our WRF modelling

experiment represents an innovative methodology for testing paleoclimate hypotheses derived from the growing number of proxy sequences available in South America.

## Code and Data availability

ERA5 Reanalysis data was retrieved from: https://cds.climate.copernicus.eu. The CHIRPS precipitation data was obtained from: https://www.chc.ucsb.edu/data/chirps. The El Niño 3.4 index was obtained from: https://climatedataguide.ucar.edu;
while the SAOD index (SAODI) was obtained from: http://lijianping.cn/dct/page/65592. Our three WRF runs (monthly climatological means), and the Python programming codes used for the climate analysis and visualizations will be freely available on GitHub at https://github.com/AlexelProgramador.

## Author contribution

IAJ, OA and AM conceptualized the study. The WRF modelling runs were performed by PS and OA. The analysis and
visualization of the data was conducted by LT-R, NL and IAJ. IAJ led the writing of the manuscript with contributions from all authors.

## Competing interests

The authors declare that they do not have any financial or personal interest that could influence the results presented in the article.

## Financial support


This research was conducted with the financial support of ANID postdoctoral grant #319018, Millennium Science Initiative Nucleus AndesPeat NCS2022_009, and Universidad de Tarapacá Mayor grant 5818-23. LT-R has received funding support from FONDECYT through grant #1201742, whereas OA was supported by FONDECYT project #11190999. The modelling work was supported by the supercomputing infrastructure of the NLHPC (ECM-02). We thank to the Center for Advanced
Studies in Arid Zones (CEAZA) and National Laboratory of High-Performance Computing Chile (NLHPC) for the availability to run the WRF simulations.

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
