# Peer review of "Exploring the causes of multicentury pluvials in the Altiplano with a climate modelling experiment"

_Climate of the Past, 2023_

## Community Comment (CC1)

Referee's main comments

**1. The abstract on its own is unclear. You should better clarify what type of model experiment(s) you perform (time period?, boundary conditions?) that lead to the changes in atmospheric circulation and sustained wet conditions (or not). Reading the abstract on its own, I was rather confused about what exactly was done in this study. For example, the result that persistent La Nina conditions inhibit moisture transport to the Altiplano is counterintuitive based on everything we know about ENSO impacts in this part of the world, and requires more context in the abstract.**

Response:

We acknowledge that the abstract should be more explicit about the specific type of model experiment we carried out. Although we focused on multi-decadal to centennial-scales trends, and therefore our experiment targeted a specific time period, our modeling study explores the effect of two contrasting oceanic-atmospheric boundary conditions on long-term hydroclimate responses. Thus, we consider our simulations to be closer to the boundary conditions type of experiment.

Although a short statement about the characteristics of our modeling experiment is included in the Abstract of the original manuscript (Lines 20-22), we acknowledge more details can be introduced to it. For this reason, we have added some additional information to the abstract, in particular:

- The regional model used
- The specific oceanic boundary conditions utilized
- The cyclical nature of the model forcing and the lack of interannual variability

Regarding the results of our modeling experiment and La Niña impacts over the hydroclimate of the Altiplano, we have added in the Abstract of the revised manuscript a direct mention about how our results contradict the present-day relationship between ENSO and the Altiplano, and how this contradiction can be relevant for interpreting paleoclimate records in the region.

**2. Discussion lines 65-70: The issue of climate models failing to reproduce observed hydroclimate changes in the region was already documented by Rojas et al. in Climate of the Past (2016). Maybe add this study as an additional example to this discussion.**

Response:

We appreciate this suggestion and, after reviewing the mentioned article, we have included a reference to the results obtained by Rojas et al. (2016) in the Introduction of the revised manuscript.

**3. Fig. 1b. I think this Figure requires a better explanation. I can't really see a clear pluvial event in these records. The actual proxies plotted should be discussed in the text or the Figure caption so it is clear what they show and how they should be interpreted.**

Response:

This is also a constructive suggestion that supports the analysis and interpretation of our results. We have added the interpretations of each of the records included in Fig. 1b in the Introduction of our revised manuscript, as suggested by the referee.

**4. The horizontal resolution of the simulation is quite course (55 km). In fact the WRF model run is coarser than the driving ERA5 dataset. Hence in effect you are upscaling and not downscaling. Why not use a 2nd embedded domain with higher horizontal resolution over the Altiplano region?**

Response:

This is a very relevant comment indeed. As the referee mentioned, our WRF model grid is indeed larger than the ERA5 forcing dataset, and therefore our experiment represents an upscaling of the lateral grid. We highlight here

that we have not used the term "downscaling" in the original manuscript, although we do acknowledge that an explicit mention of the upscaling nature of our simulations ought to be added. We have included a comment regarding the decrease in the resolution of our experiment in the Methods section of the revised manuscript.

*Why not use a 2nd embedded domain with higher horizontal resolution over the Altiplano region?*

We would like to comment that the relatively low resolution of our experiments results from the following reasons:

Firstly, as we aimed to explore the large-scale mechanisms of long-term precipitation change, it was necessary to employ a continental-scale domain, one sufficiently large to capture the modes of variability that control the moisture transport to the Altiplano from the margins of the Pacific and Atlantic ocean basins. Secondly, the requirement to simulate extraordinarily long, 100-yr periods, with boundary conditions provided at 3-hr intervals and hourly model outputs, increases significantly the computational resources (CPU time and storage) needed to conduct our simulations, preventing our ability to utilize an additional embedded domain or a lower grid size. Furthermore the current computational/processing capabilities in our national institutions impeded us from performing a high-resolution WRF simulation (<10 km) and/or a secondary domain in a reasonable time. In fact, the execution time of these simulations cannot be reduced because there is a scalability limit in relation to the number of cores used, so the waiting time for the results depends exclusively on the computing speed of the available computing nodes at the Chilean National Laboratory of High-Performance Computing (NLHPC) .

**5. You force the WRF model with ERA5 boundary conditions and then validate your WRF results with the same ERA5 dataset. That seems like circular reasoning. While the validation with CHIRPS is independent, I would not include the ERA5 comparisons unless you aim to investigate the added value from downscaling.**

Response:

We agree with the referee regarding this comment, and we have excluded the comparison with ERA5 reanalysis when discussing the validation of the WRF model in the revised manuscript. We also agree that such comparison does indeed provide important information regarding the spatial coherence of the upscaling processing. Hence, we have discussed the similarities and differences with the ERA5 dataset in the light of exploring the results the upscaling performed for our experiment.

**6. I am a bit concerned about the long-term drift in your model simulations. But it may reflect the fact that the model was forced with boundary conditions that constitute an extreme wet-year outlier. Sustaining such extreme conditions for 100 years would likely require changes in the boundary conditions that favor maintaining massive moisture transport to the Altiplano year after year. In the current environment, recharging the atmosphere with sufficient moisture is unlikely without imposing some kind of interannual climate variability that was suppressed in these simulations.**

Response:We agree with the reviewer's comments regarding the long-term drift and its possible link to the cyclic boundary conditions of the experiments. Indeed, the experimental forcing protocol suppresses interannual variability and imposes a cyclic annual forcing on both the boundary conditions and the surface data, particularly in the sea surface temperature field. However, the trend towards declining precipitation in the highlands and its increase in other regions is observed in both the experiments that replicate the conditions of extremely wet years (1983/1984 and 2011/2012) as well as in normal years (2003/2004). Based on our analyses we guess that the precipitation drift responds to progressive changes in the circulation regime that modify moisture transport and local evapotranspiration processes in particular in the Atlantic basin where the subtropical anticyclone experiment a continuous reduction that change the SACZ activity and location.

**7. Figure 4: The differences in circulation between the initial and final decade are rather hard to identify. I suggest you incorporate Fig. S9 which shows the difference between the two time periods directly into Fig. 4. It will help to highlight the actual changes. For the right column with the transect, it would make more sense to invert the color scale for humidity so more humid conditions are shown in blue and drier conditions are depicted in red, as this would be more consistent with the changes in vertical velocity depicted in the left column.**

Response:

We appreciate all these constructive suggestions made by the referee. Indeed, it becomes somehow difficult to visualize the differences between the initial and final decades of the simulation, especially for the 800 hPa circulation + specific humidity (q) diagram depicted in the center column of Fig. 4. Following the referee's suggestion, we have modified this figure, which now includes plots for the direct differences between the first and final decades of the 1983/1984 and 2011/2012 simulation periods for both:

- The 200 hPa circulation (u,v) + vertical velocity (w)
- The 800 hPa circulation (u,v) + specific humidity (q)

[Figure]

**Figure 4. Comparative panel depicting differences between the averaged atmospheric and moisture circulation during the initial and final decades of the 1983/1984 and 2011/2012 WRF simulations. (a) Difference between the final and initial decade of the 1983/1984 simulation. Left panel: vertical motion at 500 hPa (colored) along with composite upper-level (200 hPa) zonal and meridional wind differences (streamlines). Negative (positive) differences indicate increased upward (downward) motion in the final decade relative to the initial decade. Right: specific humidity at 500 hPa (colored) along with composite lower-level (800 hPa) zonal and meridional wind differences (streamlines). The red rectangle denotes the region where the longitudinal mean anomalies are calculated for Fig. 5. (b) Same for the 2011/2012 simulation.**

For the "transect" plots on the right column of Fig. 4, we have now inverted the color scale for humidity, as suggested by the referee. Unlike the 200 and 800 hPa diagrams, these "transect" plots express much more clearly the differences in circulation and moisture transport generated over the simulation's periods. Since these plots depict anomaly values relative to the entire 100-yr simulation mean, calculating the differences between the initial and final anomalies will introduce an additional (an unnecessary) layer of complexity for their interpretation. For these reasons, we have decided to keep such plots unchanged. However, since these are not direct differences like the 200 and 800 hPa diagrams of Fig. 4, we decided to include these "transect" plots in a new different figure (Fig. 5), so as not to cause any confusion between these two types of plots.

[Figure]

**Figure 5. Pressure (hPa)-latitude cross section with anomalies (z-scores) for specific humidity (colored), integrated vertical and meridional circulation (vectors), and zonal winds (contours) for the initial and final decades for the 1983/1984 and 2011/2012 simulation. The longitudinal means for all variables are calculated over the region delimited by the red rectangles in Fig. 4. Anomalies in the pressure-latitude sections were calculated as standard deviations from the 100-yr means. Continuous (dashed) contours indicate westerly (easterly) zonal wind anomalies. The red line represents the profile of maximum elevation of the Andes cordillera and the position of the Altiplano is marked. (a) and (b) First and final decade of the 1983/1984 simulation. (c) and (d) Same for first and last decades of the 2011/2012 simulation.**

**8. Figure 5: Same comment: it intuitively makes more sense to reserve blue colors for wetter and red colors for drier conditions. Hence I would suggest inverting the color scale.**

Response: We have inverted the color scales as suggested by the referee.

[Figure]

**Figure 6. Total DJF precipitation trends (mm per decade) in South America for the 1983/1984 and 2011/2012 simulations. Trends are calculated using a linear regression for the entire 100-yr simulation period.**

**9. The modeling setup is certainly interesting and the results are worth exploring. Nonetheless, it needs to be clearly stated that these are not realistic representations of boundary conditions that led to wetter conditions in the past, given that neither greenhouse gasses, volcanic, solar or orbital conditions were changed. Hence the focus here is exclusively on sustained modern wet-year boundary conditions (with a perpetual climatology and zero interannual variability), while many of the sustained changes in the past were likely externally forced. On pages 12-13 there is a long discussion about the implications of this study for paleoclimatic interpretations, e.g. during Heinrich 1, the late Holocene and the Little Ice Age. Please add a statement at the start of this discussion regarding the lack of realistic past boundary conditions and external forcings, thus limiting the ability of these simulations to serve as analogues for paleoclimatic interpretations.**

Response:

We appreciate this comment from the referee. We are aware that our experiment offers just an idealized representation and not a realistic scenario of boundary conditions leading to past hydroclimate anomalies. In agreement with the referee, we have now added a few sentences at the beginning of the "Discussion and Conclusion" section of our revised manuscript, contending that the results and interpretation should be taken with caution as our simulations do not use any realistic past boundary conditions like the ones employed in transient simulations.

Despite these real limitations, we contend that the impact of tropical Pacific or Atlantic ocean SSTs in the Holocene hydroclimate variability of the Altiplano is a matter of current debate (e.g. Jara et al., 2022; Luecke et al., 2022; Wong et al., 2021), and therefore our experiment is testing forcing mechanisms relevant to regional paleoclimatology. Although we did not alter forcing variables such as greenhouse gasses, volcanic, solar, or orbital conditions; our experiment generates useful information about the causes, mechanisms, and large-scale spatial pattern of long-term climate variability in the Altiplano and South America, which can be used to complement proxy-based reconstructions, especially records covering the mid or late-Holocene period were orbital and greenhouse parameters approached pre-industrial levels.

**10. Lines 305-311: When discussing the 2nd mode of Campos et al. (2019), please note that this study has since been updated by Orrison et al. (2022) with newer proxy data and the inclusion of an isotope-enabled model. They were able to show that the 2nd mode effectively represent SACZ variability.**

Response: We appreciate this update about the published literature. Accordingly, we have included in the revised manuscript a reference to Orrison et al. (2022) to support our argument about changes in SACZ variability.

Minor edits:

**Line 30: hypothesis => hypotheses**

Response: Done

**Line 45: pacific => Pacific**

Response: Done

**Line 53: year = years**

Response: Done

**Line 79: et al => et al.**

Response: Done

**Legend Figure 1b: polle => pollen**

Response: Done

**Legend Figure 1b: cell => cellulose**

Response: Done

**Line 101: degenerate => weaken**

Response Done

**Figure S2: above panel a) you imply that q 800 hPa is plotted, yet in the legend you write q 500 hpa => please clarify. The same applies to panel b) but here it is unclear whether omega is plotted at 200 hPa (title) or 500 hPa (caption). Finally, for vertical velocity in c) you write the unit is also m/s. Should it not be Pa/s (since you are plotting the data as a function of pressure level)? Also, please add a unit vector for vertical motion in the same way as you did for u and v. Note that some of these comments also apply to Fig S5.**

Response:

The captions of Figure S2. state "S*pecific humidity (q) at 500 hPa (shaded; g kg−1) along with integrated zonal and meridional winds at 800 hPa (streamlines)*". Hence, we disagree that we are implying the q is plotted at 800 hPa. The same applies for panel (b), as its captions clearly mentions "*Vertical motion (w) at 500 hPa (shaded; Pa s−1) and integrated zonal and meridional winds (u,v) at 200 hPa (streamlines)*". The corresponding pressure level for all variables are explicitly mentioned. For (c), we have added the integrated vertical and meridional winds unit in m s$^{-1}$, and also its approximate value in Pa s$^{-1}$.

**We also note that the unit label for the specific humidity was not added**. We have corrected this.

[Figure]

**Figure S2. 1951–2022 South America DJF climatology. (a)** Specific humidity (q) at 500 hPa (shaded; g kg−1) along with integrated zonal and meridional winds at 800 hPa (streamlines; m). **(b)** Vertical motion (w) at 500 hPa (shaded; Pa s−1) and integrated zonal and meridional winds (u,v) at 200 hPa (streamlines). Upward (downward) motion is depicted as negative blue (positive red) values. **(c)** Pressure-latitude cross section showing the integrated vertical and meridional circulation (v,w; vectors; m s−1; Pa s−1), zonal winds (u; contours), and specific humidity (q; shaded). The longitudinal means are calculated using v, w, u and q within the region delimited by the red rectangle in (a). Westerly (easterly) circulation is depicted as continuous (dashed) lines, with labels at 4 m s-1 intervals. The dashed red line of the image represents the profile of maximum elevation of the Andes cordillera along the red rectangles in (b), calculated as the maximum topographic elevation. The position of the Altiplano is labelled in the figure (c). All atmospheric data correspond to ERA5 reanalysis.

**Caption Figure S4: Climate Hazards Center InfraRed Precipitation with Station data => Climate Hazards Group InfraRed Precipitation with Station data**

Response: Done

**Caption Figure S4: CHIRPS data is plotted in orange, not green color**

Response: Caption changed accordingly

**Caption Figure S4: what do you mean with 'grilled' precipitation dataset? Maybe 'gridded'?**

Response: Yes, we meant "gridded" and we have changed accordingly.

**Table 1 and throughout the paper:The index you use is called the 'Nino3.4' index, not the 'El Nino 3.4' index. After all, this index is also used to characterize La Nina and neutral conditions in the central equatorial Pacific.**

Response: This is correct, we have amended accordingly in Table 1 and throughout the text.

**Caption Figure S5: s-1 => superscript '-1'**

Response: corrected

**Table 1: Southeastern => Southeastern**

Response: corrected

**Line 175: Climate Center for Atmospheric Research => National Center for Atmospheric Research**

Response: Done

**Line 209: parameterization => parameterization**

Response: We employed British English style in our manuscript, and therefore we prefer "parametrization". We are willing to change to the other spelling alternative in the revised manuscript if the Editor suggests so.

**Line 212: intimal => initial**

Response: Done

**Figure S7: October => October**

Response: Changed

**Line 286: accumulative => cumulative**

Response: Done

**Line 302: dryer => drier**

Response: Done

**Line 334: indicate in which journal this article was published.**

Response: We have indicated the journal as suggested

**Line 376: delete: 'J. o. G. R. A.:'**

Response: Done

**Line 377: reference is incomplete: where was this paper published?**

Response: We have added the corresponding journal.

**Line 388: you forgot to list a co-author of this paper; J. Michaelsen**

Response: We have added the author as suggested

**Line 389: reference is incomplete: where was this paper published?**

Response: We have added the corresponding journal

**Line 406: add volume or issue number or doi, so article can be traced.**

Response: Added

**Line 474: delete 'J. E. R. L.:'**

Response: Added

**Line 475: reference is incomplete: where was this paper published?**

Response: We have added the corresponding journal

**Supplement references: the Journal name where Hersbach et al (2020) was published is missing**

Response: We have added the corresponding journal

**References**

- Jara, I.A., Maldonado, A. and de Porras, M.E., 2022. Did modern precipitation drivers influence Centennial Trends in the highlands of the Atacama desert during the most recent Millennium?. Geophysical Research Letters, 49(1), p.e2021GL095927.
- Luecke, A., Kock, S., Wissel, H., Kulemeyer, J.J., Lupo, L.C., Schaebitz, F. and Schittek, K., 2022. Hydroclimatic record from an Altiplano cushion peatland (24° S) indicates large-scale reorganization of atmospheric circulation for the late Holocene. PLoS One, 17(11), p.e0277027.
- Wong, M.L., Wang, X., Latrubesse, E.M., He, S. and Bayer, M., 2021. Variations in the South Atlantic Convergence Zone over the mid-to-late Holocene inferred from speleothem $\delta18O$ in central Brazil. Quaternary Science Reviews, 270, p.107178.

---

## Author Comment (AC1)

**Response to Referee 2**

We appreciate the suggestions and open comments raised by Referee 2. We agree that our modeling experiment was not constructed with realistic paleoclimate forcing conditions, and that our paleoclimate suggestions should be taken carefully. Here we provide compelling evidence to sustain that our model simulations generate realistic long-term climate dynamics in response to the Pacific and Atlantic SSTs conditions used as external forcing. Despite the idealized nature of our WRF simulations, they provide novel and interesting insights into the causes and mechanism of long-term precipitation change in the Altiplano, a region where proxy and modeling data is still scarce. As stated in the discussion and conclusion of our submitted manuscript, such insights allowed us to assess the research questions raised in the introduction and they have the potential to contribute to the analysis and interpretation of regional paleoclimate reconstructions. Therefore, we consider our study to be worth publishing in Climate of the Past.

Below we have addressed the comments raised by the referee.

Referee #2 suggestions and open questions:

*The general setup is very artificial, since only three years (i.e. 1983, 2003 and 2011) are simulated in a perpetual mode (under present-day conditions). This setup complicates a translation back into real world conditions. Therefore statements like „To our understanding, this study presents the first regional climate modelling experiment of long-term hydroclimate responses in the southern Altiplano, yielding novel information regarding the mechanisms that govern precipitation variability at centennial timescales." is misleading, because the simulation itself is not representing a realstic 100 year period. It is just a repetition of particular years showing above normal (neutral) precipitation over the Altiplano. At least the authors should simulate a full ENSO cycle with 7-10 years and also include more information from the Atlantic Multidecadal Variability (AMV), for a more realistic simulation of the impacts of basic modes of natural climatic variability. The authors even indicate the importance of the eastern equatorial Pacific and southwestern Atlantic in their correlation map between the SSTs and the of DJF precipitation variability in the southern Altiplano (cf. Fig S3b).*

Response:

As mentioned before, we acknowledge the artificial nature of our simulations, and we agree with the referee that this should have been mentioned more clearly in the manuscript. We have now stated the idealized characteristics of our experiment in the abstract, introduction, and discussion sections of a modified version of our manuscript. Yet, the selection of the 1983/1984, 2003/2004 and 2011/2012 annual cycles to force our simulations is nor arbitrary. In section 3.1 we show that the 1983/1984 and 2011/2012 DJF seasons were both extraordinarily humid austral summers in the Altiplano (see also Fig. S4), linked to distinct, and well-defined SSTs anomalies (Fig. 2). We note that these summer seasons represent the two leading atmospheric mechanisms determining present-day interannual DJF precipitation variability in the central Andes region (Segura et al., 2020). Hence, their selection is adequate to address whether these mechanisms are able to sustain longer-term anomalies under or experimental conditions. The 2003/2004 period, on the other hand, was a neutral hydroclimate year, necessary to use as a control period to test to what extent the observed trends are developed in response to the model internal variability.

Additionally, we acknowledge that the artificial setup of our simulations implies that the paleoclimate inferences discussed in our manuscript require careful consideration. In a revised version of our manuscript, we have been explicit about these limitations at the start of the discussion section, as it was also suggested by Referee 1. Nevertheless, we contend that -despite the unrealistic nature of our experiment- our modeling results are useful to explore potential long-term precipitation trends and evaluate if the drivers of present-day humid seasons have the potential to sustain the extended pluvial events documented in published reconstructions. This information is certainly relevant for the Altiplano and for South America, where long-term climate datasets are very limited. We further maintain that we carried out an original experimental setup, identifying extraordinary hydroclimatic DJF seasons to force a detailed regional climate model. This methodology has not -to our knowledge- been employed before in this area. Hence, the sentence:

"*To our understanding, this study presents the first regional climate modelling experiment of long-term hydroclimate responses in the southern Altiplano, yielding novel information regarding the mechanisms that govern precipitation variability at centennial timescales*."

is actually not completely misleading as commented by the referee. Yet, we agree that it would be preferable to toned down the second part of the sentence to account for the uncertainties emerging for the idealize nature of our experiment:

"*To our understanding, this study presents the first regional climate modelling experiment of long-term hydroclimate responses in the southern Altiplano. Despite the idealized nature of our simulations, we have generated novel information regarding the potential mechanisms associated with precipitation variability at centennial timescale*s".

Regarding the simulation of a full ENSO cycle with 7-10 years and the inclusion of Atlantic Multidecadal Variability (AMV) for a more realistic simulation, we have acknowledged that our experiment does not introduce any source of interannual or decadal variability, and we have explicitly added this limitation to a modified version of our original manuscript. We did not want to focus exclusively on ENSO or the AMV for this manuscript, as the aim of the study was testing if specific oceanic/atmospheric conditions could generate extended periods of anomalous precipitation in our study region.

**In Fig. 3 the authors present the summertime precipitation evolution in their 100 year perpetual model simulations. All experiments, including the control experiment show a very strong decline from approx. 700 mm to 200 mm for the high precipitation events and from 500 mm to again 200 mm for the control experiment. Here I wonder why all experiments converge to the same value of 200 mm. The control experiment exactly demonstrates that the results are based on some internal model dynamics. I would expect that when the external forcing is meteorological the same at the boundaries each year, also precipitation levels should stay at least at similar (initial) levels. Therefore the question is where does all the moisture end up ? How would an experiment with considerably lower initial summertime precipitation over the Altiplano look like ? Would this also happen in reality that the Altiplano summertime precipitation would end at 200 mm ? In this context also the question remains whether the geographical domain of the regional climate model around South America is too small to account for oceanic impact (ENSO/AMV) and how/if this information is still inherited via the borders from the driving ERA5 reanalysis over the entire 100 years cycles**.

Response:

This is certainly an interesting comment that raises several issues. We will address it point by point:

**"Here I wonder why all experiments converge to the same value of 200 mm"**

Effectively, all simulations (including our control run) show negative precipitation trends, although not all of them converge to 200 mm as stated by the referee. The common drifts are acknowledged and discussed in the result sections of the original manuscript (lines 193-203). More importantly, the common hydroclimate trajectories in the Altiplano results from different mechanisms of atmospheric circulation and moisture transport. These differences indicate the existence of realistic, externally-driven long-term dynamics in each simulation, which are discussed in the manuscript and can clearly be observed in the Fig. A below, which tracks the differences in specific humidity between the start and final periods of our three simulations:

[Figure]

Figure A: Comparative panel depicting differences between the averaged specific humidity at 500 hPa (colored), along with composite lower-level (800 hPa; streamlines) zonal/meridional circulation during the initial and final decades of the WRF simulations.

Although all three simulations show significant reductions in specific humidity over the Altiplano, the 1983/1984 simulation exhibits small increments in specific humidity over the western/central Amazon and the SACZ region, whereas the other two simulations show the opposite trend for these regions. Significant differences in moisture content can also be seen in Uruguay, northern Perú and southern Brazil. As shown in Fig. 4 of our original manuscript, the drying of the Altiplano in the 2011/2012 simulation is associated with reduced humidity and upward motion in these two regions (largely imposed by cool SSTs anomalies in the subtropics and warmer anomalies in the southern Atlantic, see Fig. 2 below). By contrast, the drying in the 1983/1984 simulation is not linked to decreased humidity levels eastward from the Altiplano, but largely resulting from a prominent upper-troposphere easterly wind anomaly, as discussed in our manuscript. In sum, based on these differences, we assert that our model simulations (and the resulting precipitation trajectories) are sensitive to the distinct Pacific and Atlantic SSTs conditions used as external forcing. Hence, we disagree with the referee that our results are based on some internal model dynamics. We will provide additional evidence to refute this referee's assertion in the following responses. We further note that fig. A has been added to a modified version of the original manuscript, as suggested by Referee 1.

**"Where does all the moisture end up?"**

As shown in Fig. S8 of the original manuscript (see below), austral summer precipitation in our entire continental domain shows positive variations relative to the initial conditions over the 100-yr period in two of the three simulations (2003/2004 and 2011/2012), while the other (1984/1984) ended up with values similar to the initial state.

[Figure]

Figure S8 (original manuscript). Total annual precipitation (mm) trends for continental South America (> 10 masl) in our three modeling simulations. The shading areas correspond to the 95% confidence interval for each precipitation.

The results presented in Figure S8 imply that moisture content is not decreasing over the entire simulation domain, despite the negative trends reported for the Altiplano. Fig. 5 of the original manuscript (see below) shows that precipitation levels in both the 1983/1984 and 2011/2012 simulations increased significantly over northern Perú, Colombia, Venezuela and the northern tropical Atlantic (blue areas).

[Figure]

Figure 5 (original manuscript). Total DJF precipitation trends (mm per decade) in South America for the 1983/1984 and 2011/2012 simulations. Trends are calculated using a linear regression for the entire 100-yr simulation period.

Overall, our results show a northward moisture shift, from central Brazil and the central Andes, at the onset of the simulations, to northern South America and the northern tropical Atlantic by the last years of the experiments.

**"How would an experiment with considerably lower initial summertime precipitation over the Altiplano look like?"**

This is a very interesting point raised by the referee. We focused exclusively on positive hydroclimatic DJF seasons because our study aims to test whether the oceanic and atmospheric drivers accounting for those seasons are able to sustain extended pluvial events documented in paleoclimate sequences. However, we have actually conducted an additional 100-yr WRF simulation using the 1997/1998 DJF season as a boundary condition for a perpetual experiment. The 1997/1998 interval was a strong El Niño year and one of the driest historical austral summer seasons in the Altiplano (see Fig. S4c of the original manuscript). Figure B below shows DJF precipitation trends for the southern Altiplano in our 1997/1998 WRF simulation along with the other three experiments presented in our manuscript:

[Figure]

Figure B: Total summertime precipitation averages (mm) for the southern Altiplano region simulated in four 100-yr WRF runs. The grey shading encompasses one standard deviation from the regional averages.

Unlike the previous 3 experiments, the 1997/1998 simulations show a higher amount of precipitation at the end of the simulated period, without any downward trend as the other three simulations presented in our manuscript. This new simulation suggests that the drivers of the dry 1997/1998 DJF are able to promote an distinct internal response, with positive precipitation anomalies over the Altiplano. More importantly for this discussion, these results support our assertion that the model dynamics is influenced by external boundary conditions, and therefore the resulting long-term precipitation responses and atmospheric mechanisms are relevant for the interpretation of the paleoclimatic records. Because the results and discussion of this new 1997/1998 simulation lie beyond the scope of our study and because they are the focus of a different manuscript currently in preparation, we are inclined not to include this 1997/1998 simulation in our main manuscript or in the Supplements. However, we are open to adding this new information if the Editor requires it.

**The question remains whether the geographical domain of the regional climate model around South America is too small to account for oceanic impact (ENSO/AMV) and how/if this information is still inherited via the borders from the driving ERA5 reanalysis over the entire 100 years cycles.**

As mentioned earlier, the spatial configuration of our model effectively induces hydroclimate changes in the study area, with the magnitude and tendency of the precipitation response dependent on the SST configuration imposed in the forcing. We also note that our modeling domain was designed to maximize the oceanic component of its borders. This oceanic boundary context can be seen in Fig. 1 of our original manuscript. However, to better characterize if the domain size able to capture the Pacific and Altlantic SSTs anomalies used a boundary controls, we have added the simulation domain to Fig. 2 in a modified version of our manuscript, as seen in the right panels below:

[Figure]

Figure 2. Hydroclimate and oceanic anomalies (z-scores) during the DJF periods selected for the climate simulation experiment. (a) precipitation and SST anomalies for the 1983/1984 summer. (b) same as (a) for the 2003/2004 summer (the selected control run). (c) same as (a) for the 2011/2012 summer. All anomalies are calculated as standard deviations from the 1951-2021 mean (z-scores). The dashed gray lines on the right panels delineate the WRF simulation domain.

As seen in the figure above, the size of the simulation domain is adequate to capture the different Atlantic and Pacific SST anomalies imposed as boundary conditions.

**All experiments are carried out without any consideration of changes in orbital forcings (cf. Berger 1978), but only external forcings representing conditions of the late 20th and early 21st century. Conditions 2,000 years very different, with lower northern winter insolation and higher late northern summer/early autumn insolation, accentuating the annual insolation cycle. This might also have had an effect on the precipitation**

**dynamics but cannot be explored with the present setup. Therefore authors should use paleoclimatic Earth System model simulations representing the background conditions of their period of interest and then force their model either with a quasi-equilibrium simulation of the time period or a transient time slot. This would implicitly include i) a more realistic representation of the general external background conditions and ii) also warrant a more realistic simulation of natural climate variability that is important for comparisons with all kinds of empirical archives.**

Response:

These are again very interesting comments, which raise several issues worth discussing here. We will address them one by one:

*"Conditions 2,000 years very different, with lower northern winter insolation and higher late northern summer/early autumn insolation, accentuating the annual insolation cycle"*

As the referee commented, we did not compute any change in orbital configurations for our simulations. We consider that a more explicit mention to the unchanged nature of our external forcing (including orbital variations) would be necessary; and consequently we have added an explicit reference to this limitation at the beginning of the discussion of a new revised version of our manuscript. To visualize how much different were orbital conditions 2000 years ago, we have computed the northern (20°N) June and December insolation over the last 21,000 years, based on the Berger and Loutre (1991) orbital data table (see fig. C below).

[Figure]

Figure C. Northern hemisphere (20°N) summer and winter insolation curves according to Berger and Loutre (1991).

As pointed out by the referee, the annual insolation cycle at northern latitudes is accentuated by 2000 cal yr BP, although only by a marginal amount (difference between modern and 2000 cal yr BP seasonal variations = 2 Wm$^{-2}$) compare full range Holocene seasonal differences (e.g., difference between modern and 11,000 cal yr BP = 25.3 Wm$^{-2}$).

We can also take into consideration the Southern Hemisphere (18°S) summer insolation, which have arguably impacted the intensity of the South American monsoon over the Andes region at centennial-to-millennial timescales

(e.g., Hou et al., 2020; Ward et al., 2019). As shown Fig. D below, austral summer isolation 2000 years ago is almost identical to modern values, specially when considering the entire amplitude of Holocene variability.

[Figure]

Figure D: Southern Hemisphere (18°S) summer (21 December) insolation variability over the last 21,000 years based on the calculation of Berger and Loutre (1991).

Based on all this evidence, we disagree with the referee that orbital conditions 2000 years ago were radically different than today. These results also imply that insolation was not necessarily linked to the late Holocene pluvial events documented in the Altiplano. Hence, we contend that despite the fact that our modeling experiment did not employ past orbital parameters, our results are relevant for regional paleoclimatology.

**"Therefore authors should use paleoclimatic Earth System model simulations representing the background conditions of their period of interest and then force their model either with a quasi-equilibrium simulation of the time period or a transient time slot"**

We agree that the suggested methodology would include more realistic past boundary conditions. However, our simulation experiment aimed to explore whether some of the SSTs anomalies in the Pacific and/or Atlantic basin observed today are able to generate centennial-scale precipitation anomalies in the Altiplano, and thus performing a transient climate downscaling or a near-equilibrium simulation was beyond the scope of our study. We contend that this experimental design and the results presented in our manuscript represent a significant contribution to the paleoclimatology of South America. The following phase of our research project is to perform a dynamical downscaling of a globally-force transient climate simulation, using the WRF model with a significantly higher resolution. The results and experience of carrying out the experiment presented in our manuscript will be of great help for this and other future experiments.

**References**

- Segura, H., Espinoza, J.C., Junquas, C., Lebel, T., Vuille, M. and Garreaud, R., 2020. Recent changes in the precipitation-driving processes over the southern tropical Andes/western Amazon. Climate Dynamics, 54(5), pp.2613-2631.
- Berger, A. and Loutre, M.F., 1991. Insolation values for the climate of the last 10 million years. Quaternary science reviews, 10(4), pp.297-317.
- Hou, A., Bahr, A., Raddatz, J., Voigt, S., Greule, M., Albuquerque, A.L., Chiessi, C.M. and Friedrich, O., 2020. Insolation and greenhouse gas forcing of the South American Monsoon System across three glacial‑interglacial cycles. Geophysical Research Letters, 47(14), p.e2020GL087948.
- Ward, B.M., Wong, C.I., Novello, V.F., McGee, D., Santos, R.V., Silva, L.C., Cruz, F.W., Wang, X., Edwards, R.L. and Cheng, H., 2019. Reconstruction of Holocene coupling between the South American Monsoon System and local moisture variability from speleothem δ18O and 87Sr/86Sr records. Quaternary Science Reviews, 210, pp.51-63.